# Application of Self-Assembled Raman Spectrum-Enhanced Substrate in Detection of Dissolved Furfural in Insulating Oil

**DOI:** 10.3390/nano9010017

**Published:** 2018-12-23

**Authors:** Haiyang Shi, Weigen Chen, Fu Wan, Lingling Du, Shuhua Zhang, Weiran Zhou, Jiayi Zhang, Yingzhou Huang, Chengzhi Zhu

**Affiliations:** 1State Key Laboratory of Power Transmission Equipment & System Security and New Technology, Chongqing University, Chongqing 400044, China; weigench@cqu.edu.cn (W.C.); zhangshuhua@cqu.edu.cn (S.Z.); zhouweiran@cqu.edu.cn (W.Z.); vinisbaby@163.com (J.Z.); 2Chengdu Power Supply Company, Chengdu 610041, China; dulingling2014@163.com; 3Soft Matter and Interdisciplinary Research Center, Chongqing University, Chongqing 400044, China; yzhuang@cqu.edu.cn; 4State Grid Zhejiang Electric Power Company, Hangzhou 310000, China; yth11322@foxmail.com

**Keywords:** surface-enhanced Raman spectroscopy, transformer aging, concentration detection

## Abstract

Accurate detection of dissolved aging features in transformer oil is the key to judging the aging degree of oil-paper insulation. In this work, in order to realize in situ detection of furfural dissolved in transformer oil, silver nanoparticles were self-assembled on the surface of gold film with P-aminophenylthiophenol (PATP) as a coupling agent. Rhodamine-6G (R6G) was used as the probe molecule to test the enhancement effect. By optimizing the molecular concentration, molecular deposition time, and silver sol deposition time of PATP, the nanoparticles were made more uniform and compact, and an enhanced substrate with rich hot spots was obtained. The optimum substrate was developed, and surface-enhanced Raman spectroscopy (SERS) detection of trace furfural dissolved in transformer oil was realized. The results showed that the substrate prepared under the conditions of 0.1 mol/L PATP, 5 h deposition in PATP and 12 h immersion in silver sol, had the best reinforcement effect (that is, uniform and compact particle arrangement and no particle clusters). By use of this substrate, the minimum detectable concentration of furfural in transformer oil was about 1.06 mg/L, which provides a new method for fast and nondestructive detection of transformer aging diagnosis.

## 1. Introduction

Power transformers are indispensable equipment in a power system. Transformer failures occurring during service cause heavy economic losses and serious casualties to power suppliers. Therefore, regular diagnosis of transformer health is of great importance, especially for aging transformers [1,2,3]. The effects of temperature, electric field, water, and oxygen on the long-term operation of a transformer results in the breakdown of cellulose chains in insulating paper and produces furan derivatives dissolved in transformer oil [4,5,6,7]. It is generally believed that glucose monomers that break the ends of cellulose chains are unstable and easy to break away from cellulose chains in the process of cellulose degradation. The disintegrated glucose monomers are easily decomposed by heating, causing five furan compounds including furfural, acetyl-furan, methyl furfural, furaldehyde, and 2,5-hydroxyl-methyl-furan formaldehyde. Among them, the content of furfural is the highest [8,9,10]. Thus, it is one of the most commonly used indexes to evaluate the insulation aging of oil paper. The aging of insulating oil does not produce furfural, which is produced only by the breakage of cellulose chains in the aging process of insulating paper. Furfural in oil is determined by insulating paper. Therefore, furfural in oil exclusively reflects the aging degree of insulating paper [11,12,13,14]. This is a great advantage (furfural analysis in oil) in evaluating the aging of insulating paper. In addition, some study results have shown that furfural production is directly related to the aging of insulating paper under normal or overheated conditions compared to four other furan compounds, so furfural content could target the characteristics of the aging of insulating paper [15,16,17,18]. At the same time, another advantage of furfural detection is easy to measure. According to field transformer data, the content of furfural in oil is usually much higher than four other furan compounds, which makes it easy to accurately detect. In 1996, the content of furfural was considered to be one of the necessary testing items in document DL/T596—1996. It pointed out that the aging level of a transformer is in the middle stage of life when the concentration of furfural dissolved in oil reaches 0.5 mg/L, and the aging level of the transformer is in late life when the concentration of dissolved furfural in the oil reaches 4 mg/L [19]. Therefore, the analysis of the dissolved furfural content in transformer oil is of great significance to the correct evaluation of the aging state of a transformer. At present, the detection methods of furfural content in oil mainly include spectrophotometry and high-performance liquid chromatography [20,21,22]. These detection techniques have high detection accuracy, but they all need to extract transformer oil samples and other pretreatment. They require high requirements for operators and the detection environment, complex operation processes, and long detection cycles, which can only be completed in laboratory operations. A new detection method is an urgent need for in-site and rapid detection of dissolved furfural concentration in transformer oil. Raman spectroscopy is a spectroscopic method used to detect the vibration of molecules. The main principle of Raman spectroscopy is inelastic scattering of light irradiated on matter. In inelastic collisions, energy exchange occurs between photons and molecules. Photons not only change the direction of motion, but also transfer part of the energy to molecules. The vibrational energy of a molecule is transmitted to the photon, changing the frequency of the photon. This scattering process is called Raman scattering. Raman spectroscopy, as a single wavelength laser detection technology, has the advantages of no sample pretreatment, no loss of samples, and fast detection speed [23,24,25,26,27]. In 2015, Somekawa et al. measured the content of dissolved furfural in oil by laser Raman spectroscopy, and realized the measurement of furfural with a minimum detection concentration of 14.4 mg/L in oil [28]. In 2016, Gu et al. realized the detection of furfural in 0.1 mg/L by using confocal Raman technology and extraction technology, and achieved a maximum detection error of not more than 12.04% [29]. However, it is difficult to detect the low concentration of small molecules due to smaller molecular cross-sections. Fleischmann et al. found that pyridine molecules adsorbed on roughened Ag electrodes exhibited a large Raman scattering phenomenon [30]. In addition, the selective adsorption molecules on the active carrier surface inhibit the fluorescence emission, which greatly improves the signal-to-noise ratio of laser Raman spectroscopy. This surface enhancement effect is called surface-enhanced Raman scattering (SERS). The SERS mechanism mainly includes electromagnetic field enhancement and chemical enhancement, in which electromagnetic field enhancement is dominant. This enhancement is produced by a surface plasmon resonance effect (that is, free electrons in metals have a collective oscillation effect under the action of optical and electrical fields [31,32,33,34]). In recent years, SERS has been widely used in surface adsorption, electrochemical and catalytic reactions, chemical and biological sensors, biomedical detection, trace detection, and substance analysis [35,36,37,38,39,40,41,42]. In addition, a large number of new methods for fabricating enhanced substrates have been studied. Sergio et al. prepared uniform gold nano-octahedron structures combined with the use of a microfluidic technique based on micro-evaporation [43]. Jeong et al. prepared silver nanoshells with magnetic and SERS properties, which have been used to detect trace amounts of organic molecules [44]. Therefore, it is pretty meaningful to apply SERS technology to the detection of furfural in transformer oil.

In this paper, the effects of P-aminophenylthiophenol (PATP) concentration, deposition time in PATP, and immersion time in silver sol on the reinforcing properties of substrates were studied. The surface morphology of the substrate was characterized by X-ray photoelectron spectroscopy (XPS) and scanning electron microscopy (SEM). Rhodamine-6G (R6G) was used as the probe molecule to test the enhancement effect. By optimizing the PATP molecular concentration, PATP molecular deposition time, and silver sol deposition time, the nanoparticles were more uniform and compact, and an enhanced substrate with rich hot spots was obtained. The low concentration and in situ detection of dissolved furfural in transformer oil were realized, which provides a new method for fast and nondestructive detection of transformer aging diagnosis.

## 2. Experimental Part

Silver nitrate (AgNO_3_), P-aminophenol (PATP), sodium citrate, and Rhodamine-6G (R6G) were purchased from Aladdin (Shanghai, China). Furfural and transformer oil (Karamay 25#) were purchased from Chuan Dong chemical company of Chongqing, China. Transformer oil was used as a solvent to prepare the sample solution. In order to avoid the influence of the initial state of new oil on the experimental results, degassing and drying of the new oil were carried out before experimentation. Twenty-five milligrams of furfural were fully dissolved in 225 mL of transformer oil, and 100 mg/L standard sample solution was obtained. In order to prevent the change of mass fraction caused by the decomposition of furfural under visible light, the prepared solution was quickly sealed in a brown reagent bottle and stored in the dark. Furfural transformer oil solution of 100 mg/L could be diluted with new transformer oil proportionately to obtain furfural samples with different concentrations dissolved in the oil. The standard sample solution was diluted proportionately to obtain different concentration samples. Using the magnetron sputtering technique, a gold shell with a thickness of 100 nm was deposited on the silicon chip. Silver nanoparticles were synthesized using the sodium citrate reduction method according to Lee-Meisel [45]. The gold film was washed in water and ethanol solution for 20 min each, then immersed in PATP solution for a period of time, removed with tweezers, and washed repeatedly to remove the surface of the unbounded PATP molecules. The pretreated gold film was dried with nitrogen and soaked in silver sol for a certain time, then washed several times alternately with ethanol and deionized water and dried with nitrogen at room temperature for storage. All of the SERS experiments in this work were measured by a commercial Micro-Raman spectrometer (ANDOR, SR-5000i-C, Oxford, England), and 532 nm lasers were chosen to be the illuminating sources. The integral time of the spectrometer was set at 10 s, and the accumulated integral was 3 times. In addition, the 1200 L/mm grating and the 100-m slit width of the spectrometer were used to detect the sample. The morphology and sizes of the silver nanostructures were characterized by scanning electron microscopy (SEM, TESCAN Mira3 LMH, Brno, Czech Republic). X-ray photoelectron spectroscopy (XPS, Thermo-Fisher-Scientific ESCALAB250Xi, Shanghai, China) was used to analyze the chemical valence states of elements on the surfaces of the substrates.

## 3. Results and Discussion

Using a magnetron sputtering technique, a gold shell with a thickness of 100 nm was deposited on a silicon chip, and then PATP molecules with special functional groups were modified on the metal membrane. Finally, the prepared silver nanoparticles were deposited on the surface in order to obtain the SERS-enhanced substrate of the sandwich structure. A schematic diagram of the specific process is shown in Figure 1a. The main reason for choosing gold film as a substrate was to format a dense PATP molecular membrane on the surface. Sulfhydryl compounds and gold films can be well bonded to form S-Ag bonds. Silver nanoparticles rely on N-Ag bonds formed by adsorbed PATP functional groups, which can avoid an agglomeration effect. It was necessary to observe the deposition of silver nanoparticles on the surface of the gold film by scanning electron microscopy, considering the uneven distribution of common deposition methods. In Figure 1b, the gold film was basically covered by silver nanoparticles, which was very important for testing data stability. Due to the gold film being covered with silver nanoparticles, this indicated that adsorption of PATP molecules effectively avoided the aggregation of silver nanoparticles. It was beneficial to enhance the uniformity of the distribution of hot spots, which was particularly an important point for later experimental tests. In addition, the size and the shape of the particles had a great influence on the enhancement of the substrate. The preparation of silver nanoparticles was susceptible to the influence of temperature and external environment, which needed to be avoided as much as possible. By calculating the particle size distribution of silver nanoparticles in electron micrographs by using software (Nano Measure), it was found that the particle size was mainly concentrated in the range of 50–60 nm (Figure 1e). More importantly, in order to further determine the adsorption mode of PATP molecules on the surface of gold film, X-ray photoelectron spectroscopy tests on the modified gold film (Figure 1c, red line) were carried out. Compared to the unmodified gold film (Figure 1c, black line (Au4d_5/2_ and Au4d_3/2_ represent the presence of gold)), the characteristic lines of elemental carbons (C1s) and nitrogens (N1s) appeared, which indicated that the PATP molecules had been effectively adsorbed on the surface of the gold films. Figure 1d shows the N1s’ narrow spectrum of modified gold film. It is obvious that the characteristic peaks only appeared at 398.7 ev, which indicates that the amino groups on the PATP molecule did not react with the gold film. The reason was that new characteristic peaks appeared at 400.7 ev when the dehydrogenation of amino group occurred. This confirmed that the sulfhydryl group on the PATP molecule formed an Au-S bond with the gold film, while the amino group was far away from the gold film.

The enhancement effect mainly came from the surface plasmon resonance (SPR) produced by metal nanostructures under laser irradiation, and was mainly related to the metal material, morphology, substrate, particle size, and spacing. Previous studies have shown that the surface enhancement effects basically disappear when nanoparticle spacing is greater than 10 nm. Therefore, the substrate must satisfy the characteristics of high sensitivity and homogeneity, not only to meet the tight arrangement between particles, but also to avoid large area agglomeration effects. In the process of substrate preparation, PATP molecular concentration, molecular deposition time, and silver sol deposition time directly affect the final aggregation state of silver nanoparticles. Therefore, it was essential to further study the optimal experimental conditions. In Figure 2a, the basement and corresponding scanning electron microscope are given at PATP concentrations of 0.03, 0.05, and 0.1 mol/L, respectively. The distribution of silver nanoparticles on the substrate surface was more uniform and compact with increasing concentration. In order to evaluate the enhancement effect of the three substrates, the SERS properties of the substrates were tested for 1 × 10^−6^ mol/L R6G ethanol solution as a probe molecule. The results are shown in Figure 2a. It is seen that the prepared substrate had an obvious enhancement effect on R6G. The SERS intensity of R6G on the substrate prepared at a PATP concentration of 0.1 mol/L was the strongest, which means that the substrate prepared at a PATP concentration of 0.1 mol/L had the highest SERS activity and the best enhancement effect, and these consequences were consistent with the above characterization results. This may have been due to the strong local electromagnetic field generated when the nanoparticles were close enough to form a “hot spot”, which greatly enhanced the Raman scattering signal of the probe molecules located at the “hot spot”. In addition, the deposition time of the substrate in the solution also affected the experimental results. Figure 2b gives the SEM of five substrates prepared in the deposited solution for 1, 3, 5, 7, and 9 h, respectively. The graphs show the silver nanoparticle coverage on the surface of the gold film increased and became more uniform, from 1 to 5 h. When the substrate was deposited in solution for more than 5 h, the coverage of silver nanoparticles on the surface of the gold film increased further, but the partially agglomerated particles appeared. R6G was used as a probe molecule to test the substrate properties in Figure 2b. The results showed that the substrate enhancement effect was the best when deposited for 5 h. This may have been because with the increase of deposition time in PATP ethanol solution, the coverage of particles on the substrate surface became higher and higher, and the spacing between particles decreased, forming a large number of “hot spots”. However, with the increase in deposition time, the agglomeration of metal nanoparticles occurred, which led to a decrease in the reinforcement effect. The immersion of the substrate in solution was of great significance due to the diffusion, collision, adsorption, and binding process of silver nanoparticles under various forces. Under the action of gravity and thermodynamics, the metal particles diffused and collided, and reacted with the gold film by the adsorption of amino groups near the substrate. Thus, it was particularly important to select the best soaking time. Figure 2d shows SEM images of six substrates prepared in silver sol for 4, 8, 12, 16, 20, and 24 h, respectively. It is seen from the graph that the coverage of particles on the substrate surface under 4 h of deposition time was small, while the spacing of particles was large, so it could have been difficult to form an effective “hot spot”. With the increase of the soaking time of the substrate in silver sol, the coverage of particles on the substrate surface increased and the spacing of particles decreased, so the number of “hot spots” increased. However, as the soaking time became longer, the coverage of particles on the surface increased slowly. The main reason was that the surface coverage of particles increased as time went by during the initial adsorption. However, after a period of time, the surface coverage of the particles was almost saturated due to electrostatic repulsion between particles, so it could not be increased if the depositing time was too long. In addition, the results indicated that the enhancement effect of substrate on R6G increased with time added at a range from 4 to 12 h, but the enhancement effect was no longer obvious after 12 h. This may have been because the coverage of particles on the substrate surface was getting higher and higher, and the “hot spot” was getting richer, so the reinforcement effect on R6G was getting better and better. After 12 h of deposition, the coverage of particles on the surface was close to saturation, and the enhancement effect was barely changed. Hence, silver nanoparticles ought to have been uniformly and tightly arranged on the surface-enhanced substrate prepared by immersing in silver sol for 12 h, which had a good reinforcing effect and shortened the experimental period. According to the analysis based on the above experimental results, it could be found that there was a direct relationship between the coverage of silver nanoparticles on the substrate surface and the enhancement effect of the substrate. The aggregation state of silver nanoparticles on the substrate surface could be changed by changing the self-assembly parameters (e.g., PATP concentration, deposition time in PATP, immersion time in silver sol), and different SERS substrates with different enhanced effects could be prepared. The substrate with the best enhancement effect could be prepared by use of this method, which needs no special equipment, is simple to operate, has a low cost, and can be used to control the morphology of the substrate by various means. In subsequent experiments, we prepared the gold film as substrate for surface-enhanced Raman analysis of trace characteristics dissolved in transformer oil, under the conditions of 0.1 mol/L PATP concentration, 5 h deposition in PATP, and 12 h immersion in silver gel (More details of the substrate were included in the Appendix A).

As a kind of mineral insulating oil, transformer oil mainly consists of alkanes, cycloalkanes, and unsaturated hydrocarbons, so its Raman spectrum is more complicated and there is some fluorescence interference. As shown in Figure 3a, black lines represent part of the spectrum of transformer oil, and red lines represent the Raman spectra of the 100 mg/L furfural in transformer oil. The comparison results show that the Raman signal of furfural was completely submerged by transformer oil, and the reason was the complex composition of transformer oil. Furthermore, blue lines represent the SERS of 100 mg/L furfural concentration in transformer oil. This shows that the Raman signal of furfural could be enhanced effectively by a substrate. To further determine the Raman signal characteristic peaks of furfural on the substrate, the Raman spectra of pure furfural and substrate are given in Figure 3b. Preliminary comparisons show that the SERS spectra mainly came from the signal of the substrate itself, that is, the Raman peak of the PATP molecule. As shown in Table 1, the characteristic peaks of 1072, 1139, 1183, 1386, 1431, and 1570 cm^−1^ belonged to the characteristic peaks of the coupling molecules of PATP. The reason was that PATP was used to connect silver nanoparticles with gold film, and it was affected by electromagnetic coupling and enhanced the Raman signal. Therefore, PATP molecules appeared as strong Raman signals in the spectrum. At the same time, we found that these Raman signals overlapped severely with the Raman spectrum peaks of furfural, which brought serious interference to furfural detection. The Raman spectra of 1229, 1272, 1570, and 1662 cm^−1^ were from furfural, which corresponded to 1224, 1281, 1569, and 1670 cm^−1^ of pure furfural. This meant that the self-assembled surface-enhanced substrates on the gold film could effectively enhance the Raman signal of dissolved furfural in transformer oil, and the Raman wave number shift appeared in these four Raman spectra peaks. This was because the Raman vibration of furfural was affected by the molecule of transformer oil and the surface-enhanced substrate. In addition, the Raman-shift of these four Raman peaks was different, mainly due to physical enhancement and chemical enhancement, so the enhanced substrate had different effects toward the molecule in various modes of vibration. Among these furfural-enhanced Raman signals, the Raman peaks at 1662 cm^−1^ were enhanced, which may have been due to the fact that furfural molecules adsorbed mainly on the surface of silver nanoparticles through oxygen atoms, and the Raman peaks at 1662cm^−1^ were related to the vibration of oxygen atoms. Therefore, it could be preliminarily determined that furfural existed in transformer oil.

It was necessary to know the characteristic Raman peaks of each substance and select the corresponding characteristic Raman peaks for the accurate qualitative and quantitative analysis of each substance. According to the selection principle of Raman characteristic peaks, the characteristic Raman peaks of selected substances should not overlap with the peaks of other components and should have high intensity, and should be within the detection range of a Raman spectrometer. The Raman peaks at 1570 cm^−1^ overlapped with the base self-Raman signals. The Raman peaks at 1229, 1272, and 1662 cm^−1^ were relatively independent, but the intensity of the Raman peaks at 1662 cm^−1^ was higher. In order to obtain better detection sensitivity, 1662 cm^−1^ was selected as the characteristic peak of dissolved furfural molecule in transformer oil for further analysis. There overlapped between 1662 and 1570 cm^−1^, and Lorentzian fitting was used to separate the peaks. The goodness of fit was 0.9984, as shown in Figure 4b. A commonly used method for quantitative analysis of spectral data is the internal standard method. When using the Raman spectroscopy technique to quantitatively analyze the sample solution, the concentration difference of the sample solution, the influence of solvent noise, laser intensity, and other factors interfere with the Raman signal. The absolute peak intensity of the Raman signal fluctuates with the measurement environment, so there is a big error in quantitative analysis this way. While the internal standard substance and the sample are under the same experimental conditions, it can effectively decrease some environmental factors to use the internal standard method for quantitative analysis of the Raman spectra.

In this paper, the internal standard method was used for quantitative analysis. In SERS spectra of the sample solution, the furfural peak at 1662 cm^−1^ was taken as a quantitative peak, and the substrate peak at 1469 cm^−1^ was taken as the internal reference peak. The calibration curve equation was established by the least square fitting method, and then the substrate concentration to be measured was calculated on the basis of the ratio of the Raman peaks area. Selecting the Raman peak of substrate as the internal peak could not only avoid errors in quantitative analysis, but could reduce the interference of adding materials. After removing the baseline with baseline correction, a Savitzky–Golay polynomial was used to smooth out the noise, as shown in Figure 4a. Each Raman peak intensity at 1229, 1272, 1570, and 1662 cm^−1^ reduced with the decrease in furfural concentration. By using the least square method, linear regression was performed on the ratio of the furfural characteristic peak area at 1662 cm^-1^ to the substrate Raman peak area at 1469 cm^−1^ (A_1662 cm_^−1^/A_1469 cm_^−1^) and the concentration of dissolved furfural in transformer oil. The results are shown in Figure 4c. It is seen from the graph that there was a good linear relationship between the ratio of the Raman peak area and the concentration of furfural in the range of measured concentration, and the goodness of fit was 0.98. The linear regression equation was:

y = 0.44 x + 0.42.
(1)

The furfural concentration in transformer oil could be obtained using this equation. Therefore, the minimum concentration of dissolved furfural in transformer oil was about 1.06 mg/L according to the established detection method. After the concentration of furfural was determined, the degree of polymerization of insulating paper in transformer according to the following formula could be calculated, so the aging state of a transformer could be determined according to the following (Table 2) [46,47]:

log_10_(*C_furfural_*) = 1.56 − 0.0033*DP*(2)where *C_furfural_* is the concentration of furfural, and *DP* is the degree of polymerization of the insulating paper in transformer. Therefore, SERS technology is very promising for detecting the concentration of dissolved furfural in transformer oil and further diagnosing the aging state of a transformer.

## 4. Conclusions

In order to realize in situ detection of furfural dissolved in transformer oil, silver nanoparticles were self-assembled on the surface of gold film with P-aminophenylthiophenol (PATP) as a coupling agent. R6G was used as the probe molecule to test the enhancement effect. In the process of preparing enhanced substrates, the effects of PATP concentration, deposition time in PATP, and immersion time in silver sol on the reinforcing properties of substrates were studied. By optimizing the molecular concentration, molecular deposition time, and silver sol deposition time of PATP, the nanoparticles were made more uniform and compact, and an enhanced substrate with rich hot spots was obtained. The optimum substrate was developed, and surface-enhanced Raman spectroscopy detection of trace furfural dissolved in transformer oil was realized. The results showed that the substrate prepared under the conditions of 0.1 mol/L PATP, 5 h deposition in PATP, and 12 h immersion in silver sol had the best reinforcement effect, which was uniform and compact particle arrangement and no particle clusters. By use of this substrate, the minimum detectable concentration of furfural in transformer oil was about 1.06 mg/L, which provides a new method for fast and nondestructive detection of transformer aging diagnosis.

## Figures and Tables

**Figure 1 nanomaterials-09-00017-f001:**
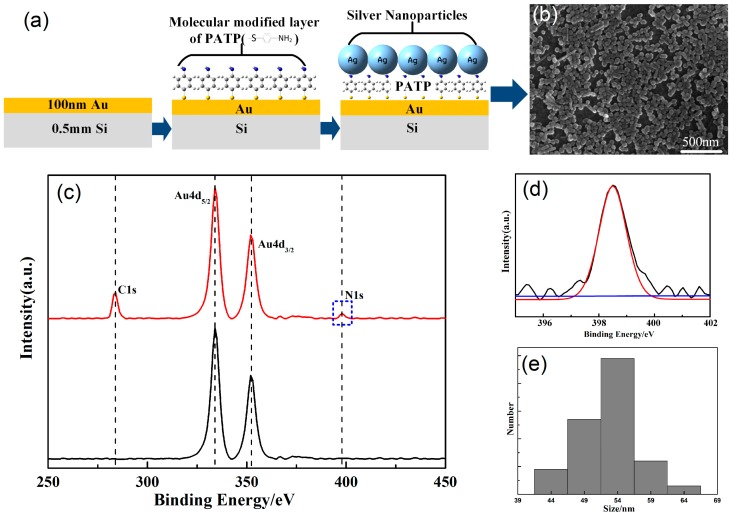
(**a**) Schematic diagram of enhanced substrate preparation; (**b**) SEM image of a surface-enhanced Raman scattering (SERS) substrate; (**c**) X-ray photoelectron spectroscopy (XPS) tests on modified gold film (red line) and no modified gold film (black line); (**d**) XPS nitrogen (N1s) spectrum for modified gold film; (**e**) size distribution of the Ag nanoparticles.

**Figure 2 nanomaterials-09-00017-f002:**
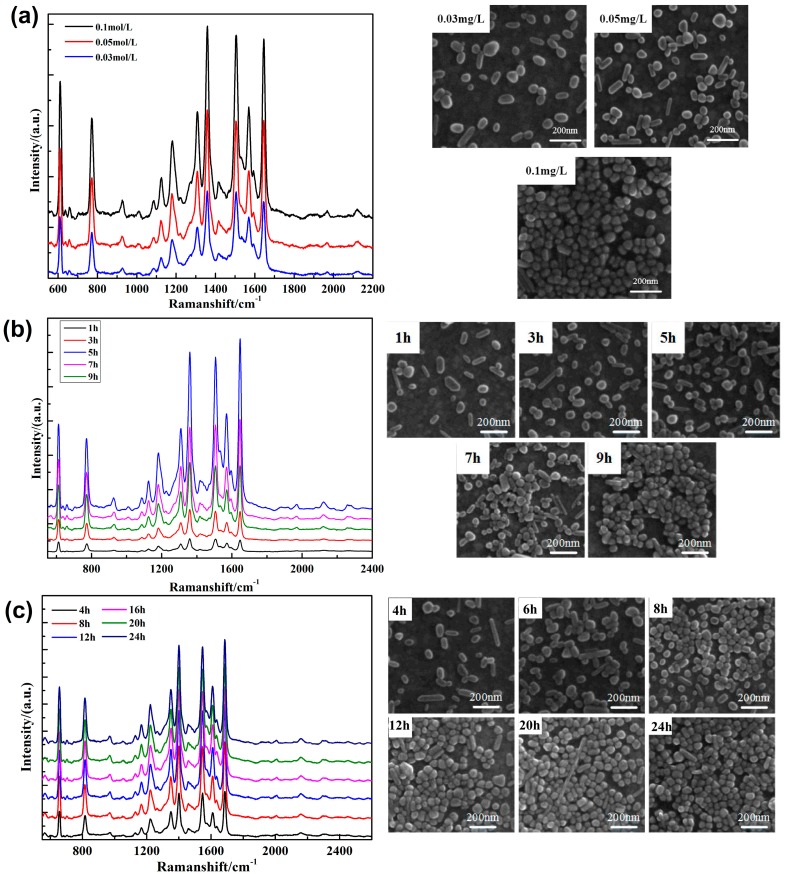
SERS spectra of 1 × 10^−6^ mol/L Rhodamine-6G (R6G) with substrates for (**a**) different P-aminophenylthiophenol (PATP) concentrations; (**b**) different deposition times in PATP; and (**c**) different immersion times in silver sol.

**Figure 3 nanomaterials-09-00017-f003:**
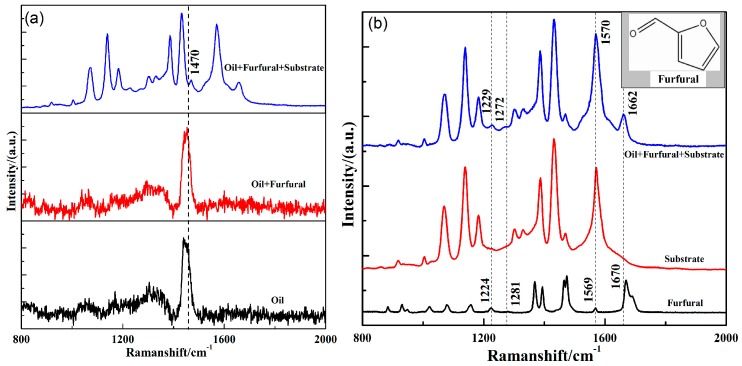
(**a**) SERS spectrum of oil with 100 mg/L furfural (Oil + Furfural + Substrate, blue line); Raman spectra of transformer oil with 100 mg/L furfural (Oil + Furfural, red line) and pure transformer oil (black line); (**b**) SERS spectrum of oil with 100 mg/L furfural (Oil + Furfural + Substrate, blue line) and substrate (Substrate, red line); Raman spectrum of pure furfural (Furfural, black line).

**Figure 4 nanomaterials-09-00017-f004:**
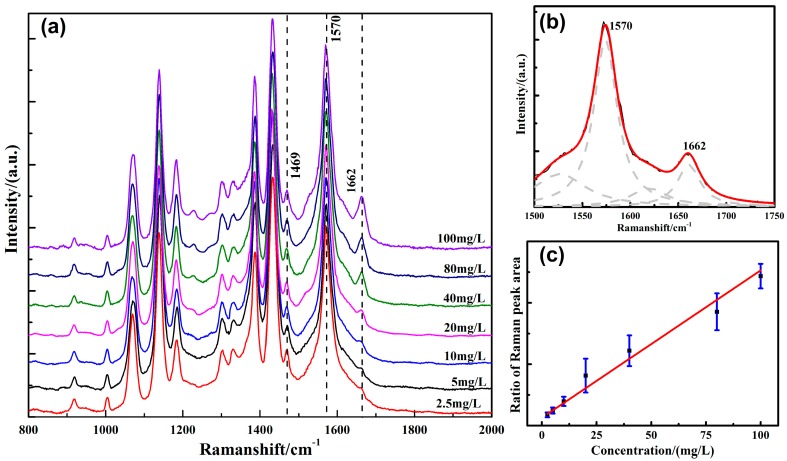
(**a**) Raman characteristic peaks of furfural at different concentrations dissolved in transformer oil; (**b**) multipeak fitting of overlapping peaks; (**c**) Raman peak area ratio (A_1662 cm_^−1^/A_1469cm_^−1^) as a function of dissolved furfural concentration.

**Table 1 nanomaterials-09-00017-t001:** Identification of characteristic peaks in SERS spectra.

Number	Ramanshift (cm^−1^)	Peak Assignment
1	919	Raman signal of substrate and transformer oil
2	1004	Raman signal of substrate and transformer oil
3	1072	Raman signal of substrate
4	1139	Raman signal of substrate
5	1183	Raman signal of substrate
6	1229	Raman signal of Furfural
7	1272	Raman signal of Furfural
8	1301	Raman signal of substrate and transformer oil
9	1334	Raman signal of substrate and transformer oil
10	1386	Raman signal of substrate
11	1431	Raman signal of substrate
12	1469	Raman signal of substrate
13	1570	Raman signal of substrate and Furfural
14	1662	Raman signal of Furfural

**Table 2 nanomaterials-09-00017-t002:** The relationship between transformer operation state and the degree of polymerization (DP) [47].

DP Value	Estimated Percentage of Remaining Life	Suggested Interpretation
800	100	Normal Ageing Rate
700	90	
600	79	
500	66	Accelerated Ageing Rate
400	50	
380	46	
360	42	
340	38	Excessive Ageing Danger Zone
320	33	
300	29	
280	24	High Risk of Failure
260	19	
240	13	End of expected life of paper
220	7	
200	0	

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
