# Peer review of "Application of Self-Assembled Raman Spectrum-Enhanced Substrate in Detection of Dissolved Furfural in Insulating Oil"

_nanomaterials, 2018, doi:10.3390/nano9010017_

Round 1

Reviewer 1 Report

This manuscript is about the preparation of SERS substrate based on the dense assembly of Ag NPs on Au film, and its application for detection of furfural in in transformer oil. Most of the presented work is focused on the optimization of SERS substrate. The title is not reflecting the work presented in the manuscript. SERS detection of  furfural is just an application. Manuscript is poorly written with many language errors. The manuscript need significant proof reading.

Comments:

1. the title should be appropriate to the work. Focus on fabrication and application to  furfural sensing in transformer oil.This give an impression of insitu detection of transformer aging by SERS, but it is not. See how insitu detection mean (Chemistry of Materials 28 (24), 9169-9180)

2. Introduction part should be revised.  There are a large number of substrates reported in literature (Chemistry of Materials 27 (24), 8310-8317, Nanomaterials 2017, 7(6), 146). What is the uniqueness of these SERS substrates.

3. Figures 1C and 1D: provide figure legends, it is hard to understand what the data correspond to.

4. Explain the data in figures 3 a and 3b properly. The figure captions are similar, so it is confusing.

5. provide molecular formula of furfural.

6. It is not clear how the SERS detection of furfural in transformer oil can probe the aging. Explain this clearly. 

7. Compare the enhancement factor of the SERS substrate with other substrates reported in literature.

Author Response

Dear editor and reviewers:

We are very thankful to the reviewers for their deep and thorough review. According to the comments, we tried our best to revise and improve the manuscript. All changes in revised manuscript are labeled in red color. And the responds to the reviewer’s comments are as follows:

Reviewer 1:

This manuscript is about the preparation of SERS substrate based on the dense assembly of Ag NPs on Au film, and its application for detection of furfural in in transformer oil. Most of the presented work is focused on the optimization of SERS substrate. The title is not reflecting the work presented in the manuscript. SERS detection of furfural is just an application. Manuscript is poorly written with many language errors. The manuscript need significant proof reading.

1. the title should be appropriate to the work. Focus on fabrication and application to furfural sensing in transformer oil. This give an impression of in-situ detection of transformer aging by SERS, but it is not. See how in-situ detection mean (Chemistry of Materials 28 (24), 9169-9180).

Response: Thanks for the reviewer’s good suggestion. The title of the article has been revised in the new manuscript. In addition, we are so sorry for our bad English writing of this manuscript. In revised one, the grammars have been improved carefully.

The corresponding changes are as follows:

“Application of Self-assembled Raman Spectrum Enhanced Substrate in Detection of Dissolved Furfural in Insulating Oil”

“ [43] Rodal-Cedeira S , Montes-García, Verónica, Polavarapu L , et al. Plasmonic Au@Pd nanorods with boosted refractive index susceptibility and SERS efficiency: A multifunctional platform for hydrogen sensing and monitoring of catalytic reactions. Chemistry of Materials, 2016, 28 (24):9169-9180.”

2. Introduction part should be revised. There are a large number of substrates reported in literature (Chemistry of Materials 27 (24), 8310-8317, Nanomaterials 2017, 7(6), 146). What is the uniqueness of these SERS substrates.

Response: Thanks for the reviewer’s good suggestion. We have supplemented the revised manuscript.

The corresponding changes are as follows:

“In addition, A large number of new methods for fabricating enhanced substrates have been studied. Sergio et al prepared uniform gold octahedron structure combined with the use of a microfluidic technique based on microevaporation.[44] Jeong et al have prepared silver nanoshell with magnetic and SERS properties, which have been used to detect trace amounts of organic molecules.[45]”

“[44] Sergio GómezGraña, Cristina FernándezLópez, Polavarapu L , et al. Gold Nanooctahedra with Tunable Size and Microfluidic-Induced 3D Assembly for Highly Uniform SERS-Active Supercrystals. Chemistry of Materials, 2015,27 (24):8310-8317.

[45] Cheolhwan J , Hyung-Mo K , So P , et al. Highly Sensitive Magnetic-SERS Dual-Function Silica Nanoprobes for Effective On-Site Organic Chemical Detection. Nanomaterials, 2017, 7(6):146.”

3. Figures 1C and 1D: provide figure legends, it is hard to understand what the data correspond to.

Response: Thanks for the reviewer’s good suggestion. We have made detailed amendments in the revised manuscript.

The corresponding changes are as follows:

X-ray photoelectron spectroscopy (XPS) test on the modified gold film (Figure 1c, red line) was carried out. Compared with the unmodified gold film (Figure 1c, black line, Au4d5/2 and Au4d3/2 represents the presence of gold), the characteristic lines of elemental carbon (C1s) and nitrogen (N1s) appeared, which indicates that PATP molecules have been effectively adsorbed on the surface of gold films. Figure 1d shows the N1s narrow spectrum of modified gold film. It is obvious that the characteristic peaks only appear at 398.7ev, which indicates that the amino groups on PATP molecule do not react with the gold film. The reason is that new characteristic peaks appear at 400.7ev when dehydrogenation of amino group occurs. It confirms that the sulfhydryl group on the PATP molecule forms Au-S bond with the gold film, while the amino group is far away from the gold film.

4. Explain the data in figures 3 a and 3b properly. The figure captions are similar, so it is confusing.

Response: Thanks for the reviewer's good question. In figure 3a, the picture shows that SERS detection can detect 100 mg/L furfural in transformer oil, but normal Raman detection cannot effectively obtain furfural signal under the same conditions. In other word, SERS has a better enhancement effect than normal Raman. However, the main purpose of Figure 3b is to accurately determine the characteristic peaks of furfural.

5. provide molecular formula of furfural.

Response: Thanks for the reviewer’s good suggestion. The molecular formula of furfural has been shown in Figure 3b in revised manuscript .

6. It is not clear how the SERS detection of furfural in transformer oil can probe the aging. Explain this clearly.

Response: Thanks for the reviewer’s good question. The corresponding changes have been given in the revised version.

The corresponding changes are as follows:

“After the concentration of furfural was determined, the degree of polymerization of insulating paper in transformer according to the following formula could be calculated, so as to the aging state of a transformer can be determined according to the following Table 2. [46-47]

Where Cfurfural is the concentration of furfural and DP is the degree of polymerization of insulating paper in transformer.

Table 2  the relation between transformer Operation State and DP

[46] Lin Y , Yang L , Liao R , et al. Effect of oil replacement on furfural analysis and aging assessment of power transformers. IEEE Transactions on Dielectrics & Electrical Insulation, 2015, 22(5):2611-2619.

[47] Dipak Mehta, Hitesh Jariwala. Predication of Life of Transformer insulation by developing Relationship between Degree of Polymerization and 2- Furfural. International Journal of Scientific & Engineering Research, 2012,3(7),1-4.”

7. Compare the enhancement factor of the SERS substrate with other substrates reported in literature.

Response: Thanks for the reviewer’s good suggestion. The enhancement effect of SERS substrate can be evaluated by enhancement factor. There are many ways to calculate the enhancement factor, and different calculation methods have different emphasis. In this paper, the calculation formulas are as follows:

where ISERS is the intensity of characteristic peaks in SERS, CSERS is the concentration of the sample in SERS, INR is the intensity of characteristic peaks in normal Raman and CNR is the concentration of the sample in normal Raman. Under the same test conditions, the intensity of 300 mg/L furfural in transformer oil is 2541.13 in normal Raman, and the intensity of 2.5 mg/L furfural in transformer oil is 72332.06 in SERS. Therefore, the enhancement factor of substrate is 3.41×103.

Reviewer 2 Report

The authors present silver nanoparticle assembled gold-film based surface enhanced Raman spectroscopy (SERS) system to detect aging of transformer oil by detecting the chemical furfural. The method of detection is novel. The manuscript might be improved by incorporating the following comments:

What is the enhancement factor for the substrate and detailed calculation should be provided to show the number of molecules detected using the SERS substrate vs. normal Raman process?

The uniformity of the substrate should be presented as a % variance in Raman intensity by performing Raman mapping over at least 100 um x 100 um area.

The mechanism of detection should be elaborated.

What is the origin for the Raman signal of substrate as generally Ag/Au does not show Raman?

In Figure 4, line 271, "(b) Raman peak area..." should be (c).

Why this particular ratio (A1662 cm-1/A1469 cm-1) as a function of dissolved furfural concentration was chosen?

In Figure 3, what's the difference between (a) and (b)?

Ideally substrate should have minimal peak. Is it the optimum substrate for the study?

Author Response

Reviewer 2:

The authors present silver nanoparticle assembled gold-film based surface enhanced Raman spectroscopy (SERS) system to detect aging of transformer oil by detecting the chemical furfural. The method of detection is novel. The manuscript might be improved by incorporating the following comments:

1. What is the enhancement factor for the substrate and detailed calculation should be provided to show the number of molecules detected using the SERS substrate vs. normal Raman process?

Response: Thanks for the reviewer’s good suggestion. The enhancement effect of SERS substrate can be evaluated by enhancement factor. There are many ways to calculate the enhancement factor, and different calculation methods have different emphasis. In this paper, the calculation formulas are as follows:

where ISERS is the intensity of characteristic peaks in SERS, CSERS is the concentration of the sample in SERS, INR is the intensity of characteristic peaks in normal Raman and CNR is the concentration of the sample in normal Raman. Under the same test conditions, the intensity of 300 mg/L furfural in transformer oil is 2541.13 in normal Raman, and the intensity of 2.5 mg/L furfural in transformer oil is 72332.06 in SERS. Therefore, the enhancement factor of substrate is 3.41×103.

2. The uniformity of the substrate should be presented as a % variance in Raman intensity by performing Raman mapping over at least 100 um x 100 um area.

Response: Thanks for the reviewer’s good suggestion, and the homogeneity of substrates is one of the important parameters of SERS. In an experiment, 20 mg/L furfural was used as probe molecule in transformer oil. Ten points were randomly selected on the same substrate to detect Raman signals, and the homogeneity of substrates was investigated in Figure 1s. The relative standard deviation (RSD) was calculated as 8.28%, which indicated that the substrate has good homogeneity.

Figure 1s

3. The mechanism of detection should be elaborated.

Response: Thanks for the reviewer’s good suggestion, and the corresponding changes have been given in the revised version.

The corresponding changes are as follows:

“And 532nm lasers were chosen to be the illuminating sources. The integral time of the spectrometer was set at 10 s, and the accumulated integral was 3 times. In addition, the 1200 L/mm grating and the 100m slit width of spectrometer were used to detect the sample.”

4. What is the origin for the Raman signal of substrate as generally Ag/Au does not show Raman?

Response: Thanks for the reviewer's good question. In this work, P-aminophenylthiophenol (PATP) molecule was used as coupling agent to assemble silver nanoparticles on the surface of gold film. Therefore, the Raman signal of the substrate mainly comes from PATP molecule in this process.

5. In Figure 4, line 271, "(b) Raman peak area..." should be (c).

Response: Thanks for the reviewer’s good suggestion, and the corresponding changes have been given in the revised version.

6. Why this particular ratio (A1662 cm-1/A1469 cm-1) as a function of dissolved furfural concentration was chosen?

Response: Thanks for the reviewer's good question. Firstly, the Raman characteristic peak of furfural is 1662 cm-1, and the characteristic peak 1469 cm-1 is the Raman signal of the PATP in SERS detection. Secondly, it is well known that the area of the characteristic peak represents the concentration of the target. Finally, in order to avoid the influence of base peaks, the internal standard method is often used for quantitative analysis of Raman. Therefore, the ratio of A41550px-1/ A36725px-1 was used as the function of the concentration of dissolved furfural.

7. In Figure 3, what's the difference between (a) and (b)?

Response: Thanks for the reviewer's good question. In figure 3a, the picture shows that SERS detection can detect 100 mg/L furfural in transformer oil, but normal Raman detection cannot effectively obtain furfural signal under the same conditions. In other word, SERS has a better enhancement effect than normal Raman. However, the main purpose of Figure 3b is to accurately determine the characteristic peaks of furfural.

8. Ideally substrate should have minimal peak. Is it the optimum substrate for the study?

Response: Thanks for the reviewer's good question. There are two unavoidable problems in the preparation of enhanced substrates. One is to try to satisfy the uniformity of the substrates and avoid the agglomeration of particles. The second is to minimize the basement signal itself.

In this article, PATP molecules with special functional groups (Sulfhydryl and amino groups) are commonly used as coupling agents. Therefore, the enhanced substrate is one of the more common methods, and we will further study better substrates in future work.

Reviewer 3 Report

Nanomaterials – 402231

The manuscript “Detection method of transformer aging state based on surface enhanced Raman spectroscopy” by Shi et al, describes the preparation of Au films coated with a linker agent and Ag NPs for the SERs detection of furfural in ageing oils. The introduction is poor and the experimental section is lack of details. The presented results are a bit confuse and the experiments should be better explained. Thus, I recommend publication after major revisions, as follows:

1. Experimental part has lack of detail. The authors should be more specific and write with detail all the experiments and characterization of the substrate. For example, how the gold film was prepared? And the Ag colloid? The time and number of acquisitions of the Raman spectra; Specify the SEM and XPS equipment; the parameter used to optimize the substrate, etc…

2. The authors claimed that the different immersion time in silver sol influence the SERWS signal of R6G, however I do not see any difference in the SERS spectra when I compare the spectra from 4h to 24h in Figure 2-c. Can the authors explain?

See other comments in attachment

Author Response

Reviewer 3:

The manuscript “Detection method of transformer aging state based on surface enhanced Raman spectroscopy” by Shi et al, describes the preparation of Au films coated with a linker agent and Ag NPs for the SERS detection of furfural in ageing oils. The introduction is poor and the experimental section is lack of details. The presented results are a bit confuse and the experiments should be better explained. Thus, I recommend publication after major revisions, as follows:

1. Experimental part has lack of detail. The authors should be more specific and write with detail all the experiments and characterization of the substrate. For example, how the gold film was prepared? And the Ag colloid? The time and number of acquisitions of the Raman spectra; Specify the SEM and XPS equipment; the parameter used to optimize the substrate, etc…

Response: Thanks for the reviewer’s good suggestion, and the corresponding changes have been given in the revised version. The result shows that the substrate prepared under the conditions of 0.1 mol/L PATP, 5 hours deposition in PATP and 12 hours immersion in silver sol has the best reinforcement effect, which has uniform and compact particle arrangement and no particle clusters.

The corresponding changes are as follows:

“Using magnetron sputtering technique, the gold shell with thickness of 100nm was deposited on the silicon chip. Silver nanoparticles was synthesized by sodium citrate reduction method according to Lee-Meisel. The gold film was washed in water and ethanol solution for 20 minutes each, then immersed in PATP solution for a period of time, and removed with tweezers, washed repeatedly to remove the surface of the unbounded PATP molecules. The pretreated gold film was dried with nitrogen and soaked in silver sol for a certain time. Then wash several times alternately with ethanol and deionized water, and dried with nitrogen at room temperature for storage. All of the SERS experiments in this work were measured by a commercial Micro-Raman spectrometer (ANDOR, SR-5000i-C), and 532nm lasers were chosen to be the illuminating sources. The integral time of the spectrometer was set at 10 s, and the accumulated integral was 3 times. In addition, the 1200 L/mm grating and the 100m slit width of spectrometer were used to detect the sample. The morphology and sizes of the silver nanostructures were characterized by scanning electron microscopy (SEM, TESCAN Mira3 LMH ). X-ray photoelectron spectroscopy (XPS, Thermo-Fisher-Scientific ESCALAB250Xi ) was used to analyze the chemical valence states of elements on the surface of substrates.”

2. The authors claimed that the different immerse on time in silver sol influence the SERWS signal of R6G, however I do not see any difference in the SERS spectra when I compare the spectra from 4h to 24h in Figure 2-c. Can the authors explain?

Response: Thanks for the reviewer's good question. The main reason for this misunderstanding is that there are too many Raman peaks and no obvious difference between them. In order to avoid this situation, the Raman spectra of 4h and 24h are shown separately in the following figure. It can be found that the effect of different time on the enhancement effect of the substrate is obvious.

3. Line 71: “measured” change to “have measured”. Line 73: “realized change to “have realized”

Response: Thanks for the reviewer’s good suggestion, and the corresponding changes have been given in the revised version.

4. what software? The authors should be more specific, at least in the experimental.

Response: Thanks for the reviewer’s good suggestion, and the corresponding changes have been given in the revised version.

The corresponding changes are as follows:

“By calculating the particle size distribution of silver nanoparticles in electron micrographs by using software (Nano Measure), it was found that the particle size was mainly concentrated in the range of 50-60 nm (Figure 1e).”

5. Figure 1a is very small. The authors should increase it. In figure 1c, each is the modified gold film and no modified gold film? the figure c is not clear.

Response: Thanks for the reviewer’s good suggestion, and the corresponding changes have been given in the revised version.

The corresponding changes are as follows:

“X-ray photoelectron spectroscopy (XPS) test on the modified gold film (Figure 1c, red line) was carried out. Compared with the unmodified gold film (Figure 1c, black line, Au4d5/2 and Au4d3/2 represents the presence of gold), the characteristic lines of elemental carbon (C1s) and nitrogen (N1s) appeared, which indicates that PATP molecules have been effectively adsorbed on the surface of gold films. Figure 1d shows the N1s narrow spectrum of modified gold film. It is obvious that the characteristic peaks only appear at 398.7ev, which indicates that the amino groups on PATP molecule do not react with the gold film. The reason is that new characteristic peaks appear at 400.7ev when dehydrogenation of amino group occurs. It confirms that the sulfhydryl group on the PATP molecule forms Au-S bond with the gold film, while the amino group is far away from the gold film.”

6. We cannot see the SEM images. They should be bigger. And the volume of Ag is the same? Or don’t have any Ag NPs? This is not clear in all manuscript?

Response: Thanks for the reviewer’s good suggestion, and the larger SEM shows the following figure at different immersion times in silver sol (4h and 24h). It is well known that the influence of external environment is inevitable in the process of substrate preparation. And the distribution of particles on the substrate cannot guaranteed to be absolutely uniform, such as red and blue square areas in the right figure. Therefore, the selection of SEM maps is to select more characteristic locations. We prefer to choose SEM with representative locations for comparison.

7. In figure 4c, error bars? How many samples did the authors test? and replicas?

Response: Thanks for the reviewer’s good suggestion, and the error bars have been given in the revised version. In addition, the homogeneity of substrates is one of the important parameters of SERS. In an experiment, 20 mg/L furfural was used as the target in transformer oil. Ten points were randomly selected on the same substrate to detect Raman signals, and the homogeneity of substrates was investigated in Figure 1s. The relative standard deviation (RSD) was calculated as 8.28%, which indicated that the prepared substrate had good homogeneity.

In order to further verify the repeatability of the experiment, the consistency of substrates prepared by self-assembly on gold film surface was studied with 20 mg/L furfural in transformer oil in Figure 2s. The RSD was calculated as 10.4%, which indicated that the substrate had good repeatability.Figure 1s SERS of furfural in transformer oil at 10 acquisition points on a single substrate

Figure 2s SERS of furfural in transformer oil on Six Substrates

Round 2

Reviewer 1 Report

The manuscript is in better shape after revision and now it can be accepted.

Reviewer 2 Report

The authors incorporated all the comments.

Reviewer 3 Report

The manuscript is better  after revision and now it can be accepted